# Seeing through Wavy Water–Air Interface: A Restoration Model for Instantaneous Images Distorted by Surface Waves

Bijian Jian [1,2], Chunbo Ma [1], Dejian Zhu [1], Yixiao Sun [1] and Jun Ao [1,*]

1   School of Information and Communication, Guilin University of Electronic Technology, Guilin 541000, China; 19021101002@mails.guet.edu.cn (B.J.); machunbo@guet.edu.cn (C.M.); 20022303185@mails.guet.edu.cn (D.Z.); 21022303047@mails.guet.edu.cn (Y.S.)
2   School of Artificial Intelligence, Hezhou University, Hezhou 542800, China
*   Correspondence: aojun@guet.edu.cn; Tel.: +86-135-5723-6811

**Abstract:** Imaging through a wavy water–air interface is challenging since light rays are bent by unknown amounts, leading to complex geometric distortions. Considering the restoration of instantaneous distorted images, this paper proposes an image recovery model via structured light projection. The algorithm is composed of two separate parts. In the first part, an algorithm for the determination of the instantaneous shape of the water surface via structured light projection is developed. Then, we synchronously recover the distorted airborne scene image through reverse ray tracing in the second part. The experimental results show that, compared with the state-of-the-art methods, the proposed method not only can overcome the influence of changes in natural illumination conditions for WAI reconstruction, but also can significantly reduce the distortion and achieve better performance.

**Keywords:** image restoration; structured light; water–air imaging; reverse ray tracing





## 1. Introduction

Viewing an airborne scene through a wavy water–air interface using a submerged camera creates a virtual periscope, which is of great significance in both the military application field and marine biology research [1–4]. Unlike other underwater imaging systems, the difficulties in such imaging scenarios mainly come from the water–air interface (WAI). Water surface fluctuations are complex and random movements, which will lead to irregular geometric distortion and motion blur in the image, causing the distortion of airborne scenes. Removing such distortions from an instantaneous distorted image is challenging since the shape of the interface is not known a priori and must be estimated simultaneously with the real scene image.

In previous works, recovering images distorted by a wavy water surface mainly relied on high-resolution video streams [5–12]. These methods always require high complexity and large data sets, so they are difficult to apply to real-time observation scenarios. However, in application scenarios such as path planning and obstacle avoidance for underwater vehicles, aerial target detection, recognition and tracking, etc., recovery methods for instant images are particularly important.

Previous studies show that the distorted scene image can be effectively recovered by reconstructing the shape of the water–air interface [13–25]. Milder et al. [15,16] emphasized first the estimation of the shape of the distorted water surface and then reconstructing the scenes. They proposed a method to estimate the water surface by analyzing the sky brightness; in their image system, an upward-looking submerged camera was used to capture the panoramic above-surface scene. The incident light from above will gradually disappear as the normal surface deviates from the viewing line of sight. Assuming that it was completely dark underwater and the sky was uniform, the brightness of the sky thus determined the surface radial slope, and they used a harmonic wave model to estimate the water surface and inverse ray tracing to reconstruct a distortion-free image.

Turalev et al. [17] studied the recovery method of water surface distortion, and then designed an experimental setup. They first utilized multiple illumination sources (red to illuminate the water surface and blue for the underwater object) to capture an image of the object and a glitter pattern of the water surface simultaneously. Then, the slope of the water surface was estimated using the glitter pattern [18]. Finally, the geometrical distortion effects of the underwater object image were eliminated by multiple accumulations of short-exposure images [19–22]. Alterman et al. [23] added an additional imaging sensor to measure the wavy water surface in real time. It works as an adaptive optics system but utilizing the sun as a guide star. The wavefront sensor consists of a pinhole array imaging the sun onto a diffuser plane directly behind it and a camera capturing the distribution of images of the sun on the diffuser plane. These pinholes have an extremely narrow field of view, so the water surface on their own line of sight is small enough to be assumed isoplanatic. Then, sampled normal vectors of the water surface were deduced from the position of each sun image through pinholes and used to estimate the water surface. Yoav Y. Schechner et al. [24] considered the problem of multi-view stereo through a dynamic refractive interface. They used multiple cameras along a wide baseline to observe a scene under uncorrelated distortions and recover sparse point clouds. R. H. Gardashov et al. [25] developed a method for recovering the single instantaneous images of underwater objects distorted by surface waves. They first determined an instantaneous shape of the wavy sea surface using the characteristics of the sun glints, and then corrected an image distorted by the wavy surface through reverse projection. However, it was only applicable to scenarios of monitoring from the sky, since there were no characteristics of sun glints when viewing the airborne scenes from underwater [15,23]. The above methods all attempted to first estimate the slope distribution of the water surface and then recovered the distorted images. However, several current methods of recovery of the water surface are not suitable for harsh application conditions, requiring special illumination settings or relying on special ambient illumination conditions (requiring the sun or uniform brightness in the sky). Furthermore, the accuracy of the estimation is also unable to meet the requirements.

In order to overcome the dependence of previous methods on natural illumination conditions and realize the distortion correction of instantaneously distorted images, an image restoration model based on structured light projection is proposed in this paper. Compared to previous approaches, our method does not require special natural illumination [15–17,23,25], multiple viewpoints [24] or a complex experimental setup [17,23]. It only requires a simple projection setup and an image of the distorted scene. The main contributions of this paper are as follows: (1) we propose a new image restoration model for instant images via structured light projection, (2) we introduce a WAI reconstruction algorithm based on structured light, (3) we analyze some of its limitations.

## 2. Optical Analysis of Imaging through Refractive Media

### 2.1. Snell's Window

By imaging through the water–air interface using a submerged camera, one can observe the whole sky. However, it does not stretch 180° from horizon to horizon, as it does above water. Instead, it is compressed into a circular area spanning approximately 97.2°, regardless of the observer's depth. According to Snell's Law [26,27], this occurs because light rays are bent when entering or exiting water. The shrunken sky (celestial hemisphere) seen by submerged observers is called Snell's window (SW). SW is surrounded by a dark field that represents light that is entirely internally reflected from the sea and back to the observer from the underside of the water's surface.

As shown in Figure 1, a submerged camera observes from underwater at 3D location C. CM and CN are the total reflection boundaries, ∠MCN = 97.2°. The conical area formed by the rotation of CM and CN around the optical axis is the Snell cone, and the circular area on the water surface between them is Snell's window, the boundary of which is the extinction boundary, and the outside is the dark field.

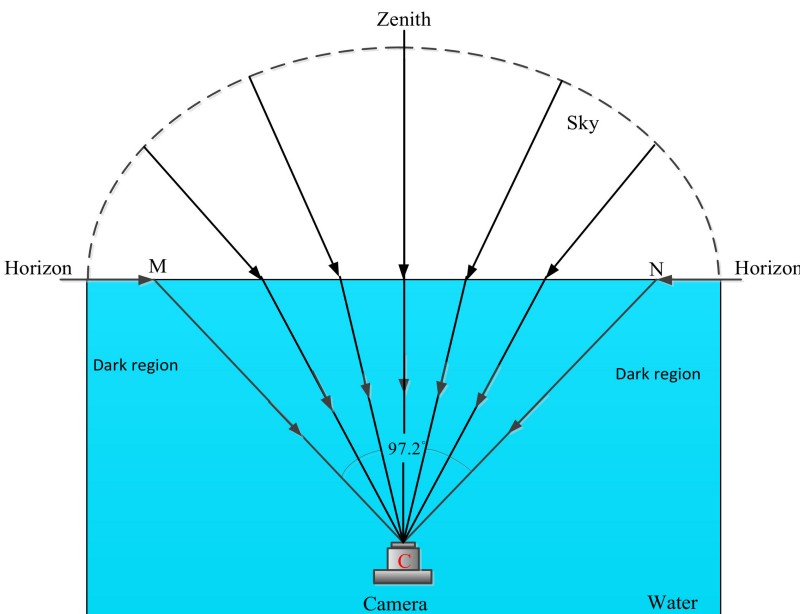

**Figure 1.** Optics of Snell's window for flat water.

Assuming that the brightness value of the hemisphere space above the water surface is 1, the brightness value of the underwater scene is 0, and the energy loss of absorption and scattering by the water body is ignored. Figure 2 shows the normalized illuminance as a function of the height angle of incident light rays (blue) and the normalized illuminance as a function of the elevation angle of the field of view of the camera (red) when the water surface is still. According to Figure 2, it does not receive the irradiance from the sky at all when the elevation angle of the camera $\theta_{water} \in (0°, 41.4°) \cup (138.6°, 180°)$. This occurs because, within this field of view, light rays are totally internally reflected from the sea and back to the observer from the underside of the water's surface. In deep water, there is very little light coming from below and so this part is dark and shows no apparent color or structure. Therefore, the field of view of the underwater camera is constricted within the Snell window within $\theta_{water} \in (41.4°, 138.6°)$.

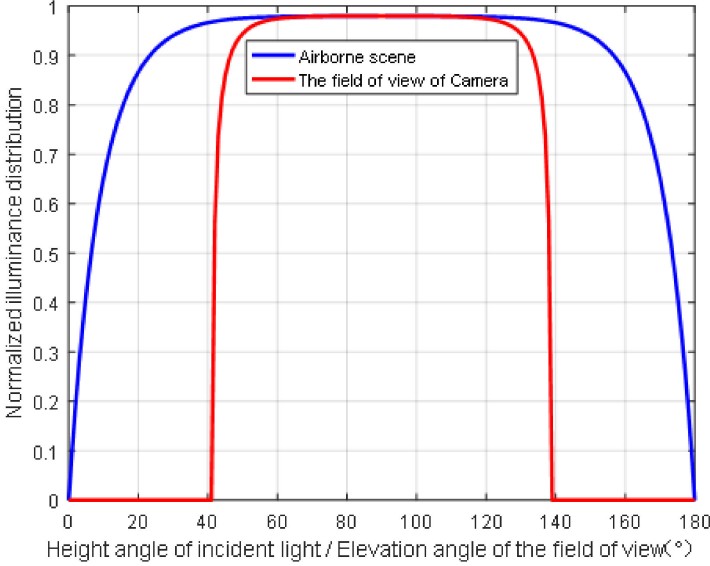

**Figure 2.** Normalized illuminance distribution of Snell's window for flat water.

### 2.2. Optical Properties of Sea Water

The energy attenuation of light in water is mainly caused by the absorption attenuation of the water body and the scattering effect of suspended particles in the water. Studies have shown [28–30] that the transmission of light in water is caused by two independent physical processes, namely absorption and scattering, and the energy decays exponentially. According to the Beer–Lambert law, the radiance attenuation of monochromatic light can be expressed as

$$I_d = I_0 \, e^{-c(\lambda_i) \cdot d}, \tag{1}$$

where $I_d$ is the intensity of a radiation of wavelength $\lambda_i$ with the starting intensity $I_0$ after traveling the distance $d$ through a material with the attenuation coefficient $c(\lambda_i)$. The attenuation coefficient $c(\lambda_i)$ consists of the energy losses due to the absorption coefficient $a(\lambda_i)$ and the scattering coefficient $b(\lambda_i)$:

$$c\,(\lambda_i) = a(\lambda_i) + b(\lambda_i). \tag{2}$$

C. Smith et al. [31] performed a detailed measurement of the attenuation coefficient of light in clear seawater and gave the measurement data of the absorption attenuation coefficient and scattering coefficient of seawater as a function of wavelength. The absorption and scattering attenuation coefficients of seawater as a function of wavelength are shown in Figure 3. The results show that seawater displays the selective absorption of light of different wavelengths, in addition to the existence of scattering properties. As shown in Figure 3, it is not difficult to find that the transmittance of seawater in the blue–green band of the spectrum is relatively large, and the light energy attenuation is the smallest, which is called the "blue–green window". Therefore, in structured light projection, a light source in this band can be selected for wavefront sampling to reduce the influence of light absorption and scattering on the WAI reconstruction.

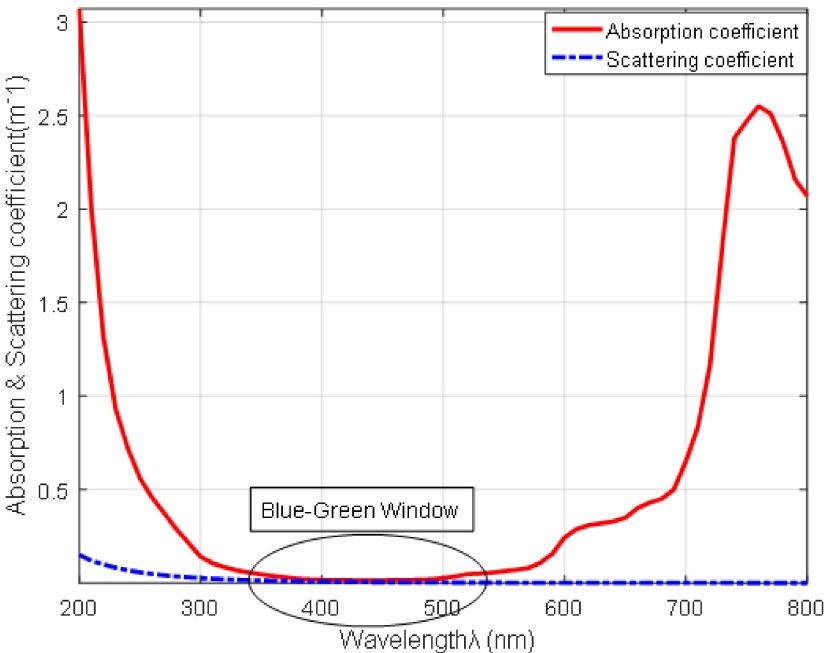

**Figure 3.** Absorption and scattering coefficients of light in pure ocean.

## 3. Materials and Methods

### 3.1. Model Descriptions

The system model is shown in Figure 4, in which the projector first projects an adaptive and adjustable structured light pattern onto the water surface, and then the camera s acquires the distorted structured light image from the diffuser plane, while the camera

v captures the airborne scene through the same WAI. The algorithm is composed of two separate parts. In the first part, an algorithm for the determination of the instantaneous shape of the water surface from structured light is developed. Then, we synchronously recover the distorted airborne scene image through reverse ray tracing in the second part.

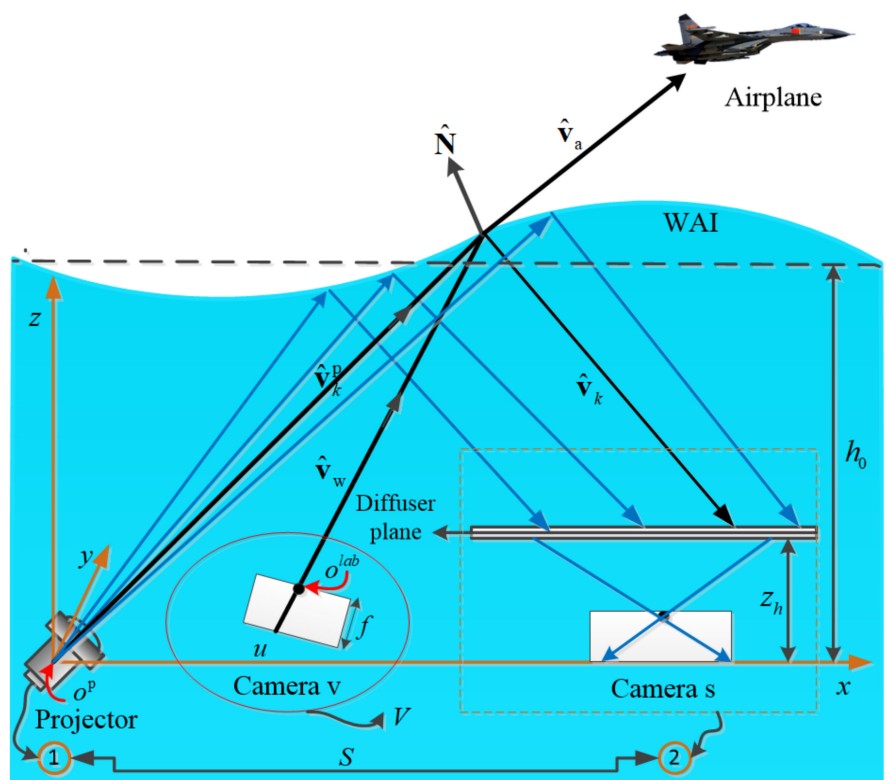

**Figure 4.** Geometry of the image restoration model via structured light projection, comprising a structured light projection system $S$ and an observing system $V$, where component $S$ includes a projector, diffuser plane and camera.

### 3.2. WAI Reconstruction Algorithm Based on Finite Difference

In this section, we propose an algorithm for the determination of the instantaneous shape of the water surface from structured light. According to the law of reflection, we first calculate the WAI normals of sampled points using the location information of the feature points between the reference structured light image and the distorted structured light image. Then, the WAI shape is estimated utilizing the finite difference method.

#### 3.2.1. Sampling of WAI Normals

The algorithm first takes advantage of the position information of the feature points of the structured light pattern to perform the quasi-periodic sampling of the wave surface to be measured. Figure 5 shows a simulation example of WAI sampling via structured light projection. Figure 5a is the preset structured light pattern, and Figure 5b is the projection on the water–air interface when the water surface is flat (note: in the daytime or moonlit night, no projection is formed on the water–air interface; a virtual image is introduced for the convenience of analysis). Given the system parameters, the location distribution of the WAI sampling points can be obtained using perspective projection transformation [32–34], as shown in Figure 5c.

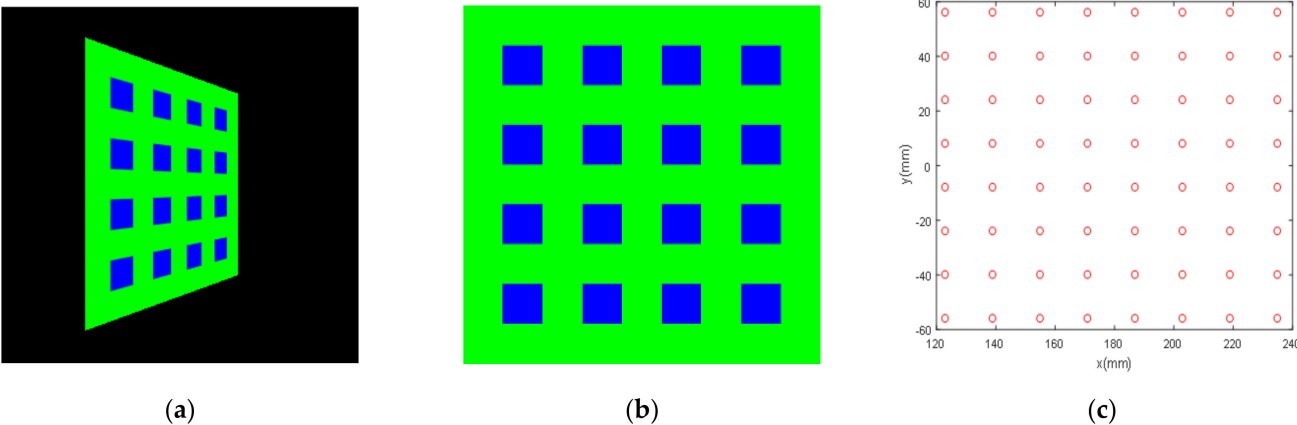

(**a**)                                                                 (**b**)                                                                 (**c**)

**Figure 5.** Examples of WAI sampling via structured light projection: (**a**) structured light pattern on the display element of the projector; (**b**) the virtual image formed on the water surface via structured light projection when the WAI is flat; (**c**) sampling point distribution of the WAI for flat water surface.

The structured light pattern of blue–green stripes projected by the projector is reflected by the water surface, forming a distorted structured light image on the diffuser plane. This section solves the sampled normals of the WAI according to the relationship between the incident ray, the reflected ray and the normal vector.

As shown in Figure 6, the global coordinate system is established with the projected center $\mathbf{o}^{\text{pro}}$ as the origin, and the $z$ axis denotes the height above the projector. $\mathbf{p}_k$ is the 3D location of the arbitrary feature point on the structured light pattern to be projected, and the corresponding projected ray is reflected by the WAI, at 3D location $\mathbf{q}_k$, and then the reflected ray irradiates a spot on the diffuser $\prod^{\text{difusser}}$, at 3D location. $\hat{\mathbf{N}}_k$ is the corresponding WAI normal, and $\mathbf{s}'_k$ is the corresponding spot when the WAI is flat.

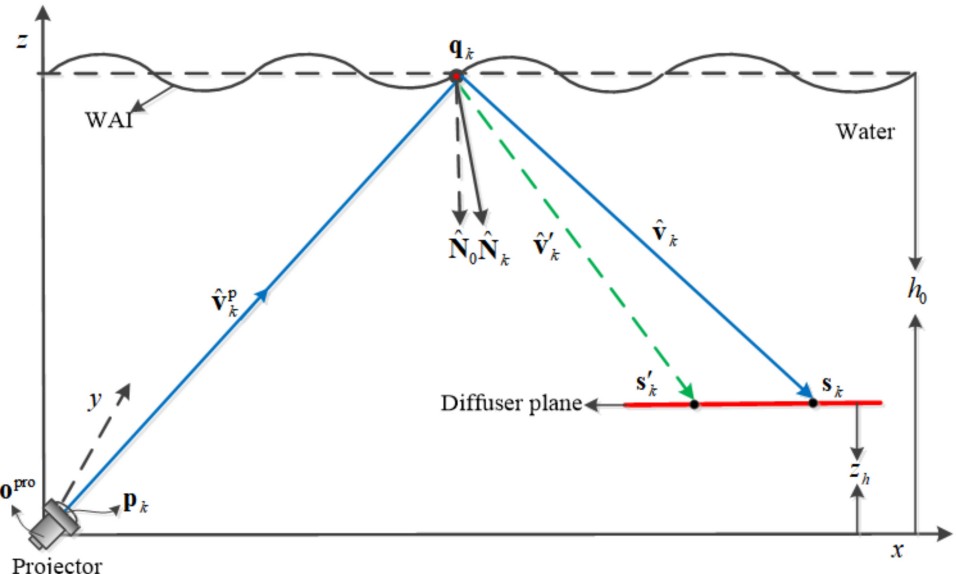

**Figure 6.** The principle of sampling of WAI normals.

The unit vector of projected ray $\hat{\mathbf{v}}_k^{\text{p}}$ is always known, given the preset structured light pattern. As shown in Figure 5c, the set $\{\mathbf{q}_k\}$ represented as $\{\mathbf{q}'_k\}$ is periodic when the WAI is still. Meanwhile, $\{\mathbf{q}_k\}$ is quasi-periodic, having a perturbation to periodicity, when the WAI is wavy. Considering that the variations in the height of the water surface are small compared to the work depth of the system ($\Delta h \ll h_0$), we have

$$\mathbf{q}_k \approx (q_x, q_y, h_0) = \mathbf{p}_k + (h_0/c_k^{\text{P}}) \cdot \hat{\mathbf{v}}_k^{\text{P}}, \tag{3}$$

where $h_0$ is the system height, which is the average underwater depth of the projector. The value can be determined in the field, using a pressure-based depth gauge. $c_k^P$ is the propagation length of $\hat{\mathbf{v}}_k^P$ at the z-axis. Moreover, the 3D location of the spot $\mathbf{s}_k$ can be extracted from the distorted structured light image. We adopt the corner detection algorithm of Reference [35] for feature extraction and matching. Therefore, the vector of reflected ray $\hat{\mathbf{v}}_k$ is $\hat{\mathbf{v}}_k = \mathbf{s}_k - \mathbf{q}_k / \|\mathbf{s}_k - \mathbf{q}_k\|$. Using the vector form of the law of reflection at the water interface,

$$\hat{\mathbf{v}}_k \times \hat{\mathbf{N}}_k = \hat{\mathbf{v}}_k^P \times \hat{\mathbf{N}}_k \Rightarrow \left( \hat{\mathbf{v}}_k^P - \hat{\mathbf{v}}_k \right) \times \hat{\mathbf{N}}_k = 0. \tag{4}$$

Here, $\times$ is the cross product. Using the axial components of the vectors $\hat{\mathbf{v}}_k^P$, $\hat{\mathbf{v}}_k$ and $\hat{\mathbf{N}}_k$, Equation (4) can be converted into the dot product form,

$$\begin{bmatrix} 0 & -c_k^P + c_k & b_k^P - b_k \\ c_k^P - c_k & 0 & -a_k^P + a_k \\ -b_k^P + b_k & a_k^P - a_k & 0 \end{bmatrix} \cdot \begin{bmatrix} N_x \\ N_y \\ N_z \end{bmatrix} = 0, \tag{5}$$

where $a_k^P, b_k^P, c_k^P$ are the $x$, $y$ and $z$ components of the vector $\hat{\mathbf{v}}_k^P$, respectively. $a_k, b_k, c_k$ are the axial components of the vector $\hat{\mathbf{v}}_k$. $N_x$, $N_y$, $N_z$ are the axial components of the vector $\hat{\mathbf{N}}_k$. Equation (5) also can be phrased in matrix form

$$\mathbf{A}\hat{\mathbf{N}}_k = 0, \tag{6}$$

where

$$\mathbf{A} = \begin{bmatrix} 0 & -c_k^P + c_k & b_k^P - b_k \\ c_k^P - c_k & 0 & -a_k^P + a_k \\ -b_k^P + b_k & a_k^P - a_k & 0 \end{bmatrix}. \tag{7}$$

The WAI normal $\hat{\mathbf{N}}_k$ is estimated by solving Equation (6): it is the null subspace of $\mathbf{A}$. This process is repeated for each sampled point located at $\mathbf{q}_k$. It yields a set of sampled vectors $\{\hat{\mathbf{N}}_k\}$ corresponding to the set $\{\mathbf{q}_k\}$.

3.2.2. Reconstruction of the WAI

Assume that $h(x, y)$ is the height of an arbitrary sampled point $(x, y)$ of the water surface, and $\mathbf{Z}$ is a 2D height field. The gradient of $\mathbf{Z}$ can be given by

$$\nabla \mathbf{Z} = \frac{\partial h(x, y)}{\partial x} \hat{\mathbf{i}} + \frac{\partial h(x, y)}{\partial y} \hat{\mathbf{j}} = \mathbf{Z}_x(x, y) \hat{\mathbf{i}} + \mathbf{Z}_y(x, y) \hat{\mathbf{j}}, \tag{8}$$

where $\mathbf{Z}_x(x, y)$, $\mathbf{Z}_y(x, y)$ are the $x$ and $y$ components of the two-dimensional numerical gradient of the arbitrary sampled point $(x, y)$, respectively. $\hat{\mathbf{i}}$ and $\hat{\mathbf{j}}$ are the unit vectors in the $x$, $y$ axis directions, respectively. Therefore, we can obtain the first-order partial differential equation as follows:

$$\begin{cases} \frac{\partial h(x,y)}{\partial x} = \mathbf{Z}_x(x, y) \\ \frac{\partial h(x,y)}{\partial y} = \mathbf{Z}_y(x, y) \end{cases}, \tag{9}$$

The sampled normals $\{\hat{\mathbf{N}}_k\}$ corresponding to the sampled points $\{\mathbf{q}_k\}$, estimated in Section 3.2.1, are known. Moreover, the normal vector of an arbitrary sampled point on the wave surface can be expressed as $[\mathbf{Z}_x(x, y), \mathbf{Z}_y(x, y), -1]$. For the gradient operator of the 2D discrete function $h(x, y)$, according to the finite difference theory [36], we use the central difference formula to approximate the first derivative. Therefore, Equation (9) becomes

$$\begin{aligned} \frac{\partial h(x,y)}{\partial x} &= \frac{h(x_{i+1}, y_j) - h(x_{i-1}, y_j)}{\Delta x_i + \Delta x_{i-1}} & i = 1, 2 \ldots n; \ \Delta x_i = x_{i+1} - x_i; \ \Delta x_{i-1} = x_i - x_{i-1} \\ \frac{\partial h(x,y)}{\partial y} &= \frac{h(x_i, y_{j+1}) - h(x_i, y_{j-1})}{\Delta y_j + \Delta y_{j-1}} & j = 1, 2 \ldots m; \ \Delta y_i = y_{i+1} - y_i; \ \Delta y_{i-1} = y_i - y_{i-1} \end{aligned}, \tag{10}$$

where $m, n$ represent the dimensions of the grid of sampled points $\{\hat{\mathbf{N}}_k\}$. Noting that **H** is the vector of length mn representing the height field $h\,(x, y)$ sampled on a $m \times n$ grid, the two vectors of length mn representing each component of the gradient field can be written as

$$\begin{aligned}\mathbf{G}_x \cdot \mathbf{H} &= \mathbf{Z}_x \\ \mathbf{G}_y \cdot \mathbf{H} &= \mathbf{Z}_y\end{aligned} \tag{11}$$

where $\mathbf{G}_x$ and $\mathbf{G}_y$ are two sparse matrices, of size mn $\times$ mn, defining the linear combinations of the elements of **H** to produce each gradient. Equation (11) can be merged into a single linear system,

$$\mathbf{G} \cdot \mathbf{H} = [\mathbf{Z}_x, \mathbf{Z}_y]^{\mathrm{T}} = \Xi, \tag{12}$$

where $\mathbf{G} = [\mathbf{G}_x, \mathbf{G}_y]^T$ is a rectangular sparse matrix of size 2mn $\times$ mn, and $\Xi$ is a vector of length 2mn. This system thus gives 2mn equations with mn unknowns. It is overdetermined, so a direct inversion is not possible. However, an estimate of **H** may be obtained, by minimizing the residual [37],

$$\|\mathbf{G} \cdot \mathbf{H}\text{-}\Xi\|^2 \to \min, \tag{13}$$

where $\| \cdot \|$ represents the Euclidean norm. The WAI shape estimated by the numerical integration of the inverse gradient operator is sparse and discontinuous, whereas the WAI is typically smooth and integrable. Thus, we further perform the bicubic interpolation algorithm [38] for **H** to estimate the WAI shape.

### 3.3. Image Restoration Algorithm through Ray Tracing

Component $V$ of the imaging sensor views the airborne scene through the wavy WAI, as shown in Figure 4. As the shape of the WAI is known, according to the principle of 3D camera imaging, this section proposes an image restoration algorithm based on inverse ray tracing. The principle of the algorithm is shown in Figure 7.

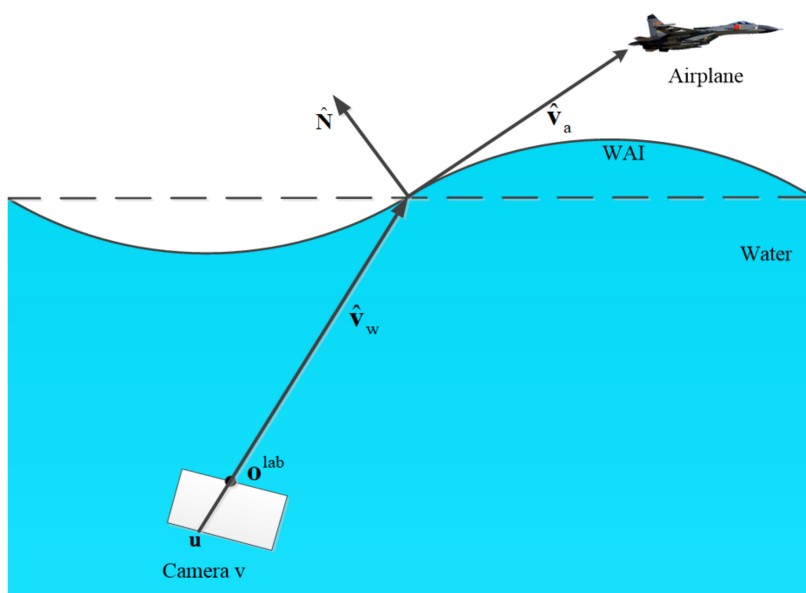

**Figure 7.** The principle of reverse ray tracing.

An internal coordinate system of $V$ consists of its optical axis and the lateral pixel coordinates of the image plane. The origin of the coordinate system of camera v is at the optical center $\mathbf{o}^{\text{lab}}$. The optical axis intersects the image plane at location **c**. The focal length of camera v is $f$. In this coordinate frame, the 3D location of the pixel **u** is

$$\mathbf{u}^{\text{cam}} = [\mathbf{u}^{\mathrm{T}}, -f]^{\mathrm{T}}, \tag{14}$$

where **u** is the 2D location of the pixel on the image plane, and T represents transposition. Relative to the global coordinate system, the pose of camera v is defined by a rotation matrix **R** and a translation vector **t**. In the global coordinate system, the 3D location of the pixel **u** can be expressed as

$$\mathbf{u}^{\text{lab}} = \mathbf{R}^{\text{T}}(\mathbf{u}^{\text{cam}} - \mathbf{t}). \tag{15}$$

Let $\mathbf{u}^{\text{cam}} = 0$ in Equation (15), and the origin of camera v in the global coordinate system is

$$\mathbf{o}^{\text{lab}} = -\mathbf{R}^{\text{T}}\mathbf{t}. \tag{16}$$

In the global system, according to inverse ray tracing [31], the back-projected ray from $\mathbf{u}^{\text{lab}}$ through $\mathbf{o}^{\text{lab}}$ can be given by

$$I^{\text{w}}(\mathbf{u}) \equiv \left\{ \mathbf{o}^{\text{lab}} + \hat{\mathbf{v}}_{\text{w}} l \right\}_{\forall l > 0}, \tag{17}$$

where the ray direction vector is

$$\hat{\mathbf{v}}_{\text{w}} = (\mathbf{u}^{\text{lab}} - \mathbf{o}^{\text{lab}}) / \left\| \mathbf{u}^{\text{lab}} - \mathbf{o}^{\text{lab}} \right\|, \tag{18}$$

where $l$ denotes the propagation length along the ray. In a perspective system, the inverse projected ray intersects the water–air interface at

$$\mathbf{q}(\mathbf{u}) = \text{WAI} \cap I^{\text{w}}(\mathbf{u}), \tag{19}$$

where the WAI estimated in Section 3.2 is known. The WAI normal vector is $\hat{\mathbf{N}}$. According to the vector form of Snell's law, the direction vector of the airborne observing ray is given by

$$\hat{\mathbf{v}}_{\text{a}} = n\hat{\mathbf{v}}_{\text{w}} + \hat{\mathbf{N}} \left[ \sqrt{1 - n^2 + n^2(\hat{\mathbf{v}}_{\text{w}} \cdot \hat{\mathbf{N}})^2} - n\hat{\mathbf{v}}_{\text{w}} \cdot \hat{\mathbf{N}} \right], \tag{20}$$

where $n$ is the refractive index of water, $n = 4/3$. $\hat{\mathbf{v}}_{\text{a}}$ is the undeflected direction vector pointing to the object. Supposing that the airborne rays $\{\hat{\mathbf{v}}_{\text{a}}\}$ of all of the pixels in the distorted image intersect with a plane in the air $\prod^{\text{object}}$, we can easily recover the distorted image through perspective projection.

## 4. Limitations

The limitations of system component $V$ are described in detail in Ref. [23]. Hence, we focus here on the limitations, sensitivities and resolution trade-off concerning only the structured light projection system $S$. Primarily, $S$ is not limited by natural illumination conditions. It is suitable for daytime as well as moonlit nights. Other sensitivities and limitations of $S$ are geometric, as analyzed next.

### 4.1. Sensitivity to Variations in $\hat{N}$ for Structured Light

The WAI normal is perturbed around the $z$ axis. The direction vector of the reflected ray $\hat{\mathbf{v}}$ is obtained by the law of reflection,

$$\hat{\mathbf{v}} = \hat{\mathbf{v}}^{\text{P}} - 2\hat{\mathbf{N}} (\hat{\mathbf{N}} \cdot \hat{\mathbf{v}}^{\text{P}}), \tag{21}$$

where $\hat{\mathbf{v}}^{\text{P}}$ is the direction vector of the projected ray from the projected center through an arbitrary feature point. Substituting $\hat{\mathbf{N}} = -\hat{\mathbf{z}}$ into Equation (21) yields the direction vector of the reflected ray, when the WAI is flat, $\hat{\mathbf{v}}'$. When the water is wavy, the reflected angle changes by

$$\psi = \arccos(\hat{\mathbf{v}} \cdot \hat{\mathbf{v}}') \tag{22}$$

The perturbation has two principal components. The first principal component is the meridional plane component, which is in the *xoz* plane. In this component, the WAI normal $\hat{\mathbf{N}}$ rotates around the $y$ axis. Here, we analyze the sensitivity of structured light to changes in $\hat{\mathbf{N}}$ by taking the projected ray along the optical axis of the system as an example. Assuming that $\theta_{\text{pro}}$ is the elevation angle of the projector, and the unit vector $\hat{\mathbf{v}}^{\text{P}} = \hat{\mathbf{v}}^{\text{P}}_{\text{oPro}} = [\cos\theta_{\text{pro}}, 0, \sin\theta_{\text{pro}}]^{\text{T}}$, the corresponding normal is

$$\hat{\mathbf{N}} = \hat{\mathbf{N}}_{\text{xoz}} = (\; \sin\theta \;,\; 0\;,\; -\cos\theta\;)^{\text{T}}, \tag{23}$$

where $\theta$ is the inclination angle of the WAI. Another perturbation component is the sagittal plane, which resides in the *yoz* plane. In this component, $\hat{\mathbf{N}}$ rotates around the $x$ axis; then,

$$\hat{\mathbf{N}} = \hat{\mathbf{N}}_{\text{yoz}} = (\; 0, \sin\theta, -\cos\theta\;)^{T}, \tag{24}$$

Substituting Equations (23) or (24) into Equations (21) and (22), we derive the angular perturbation of the reflected ray. Sensitivity to perturbations is assessed by $d\psi/d\theta$. Figure 8 shows the angle of deflection of the reflected ray as a function of the inclination of the WAI when $\theta_{\text{pro}} = 30°,\; 45°$, respectively. The results reveal that the sensitivity of the structured light to the variations in the WAI normal is a constant value, namely $d\psi/d\theta = 2$.

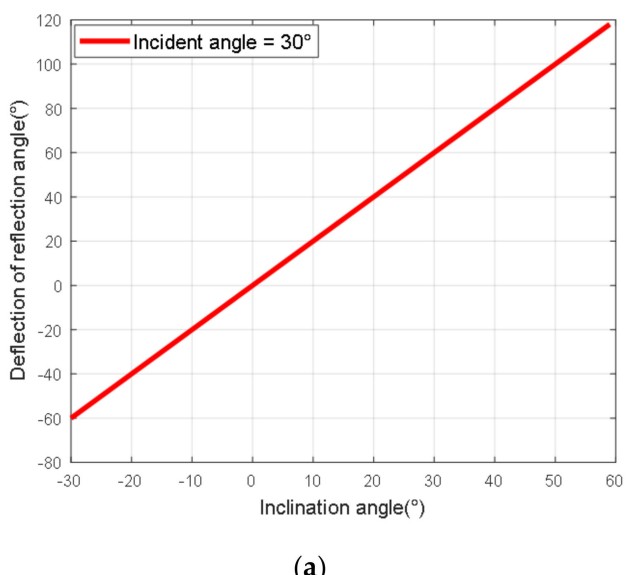

(**a**)

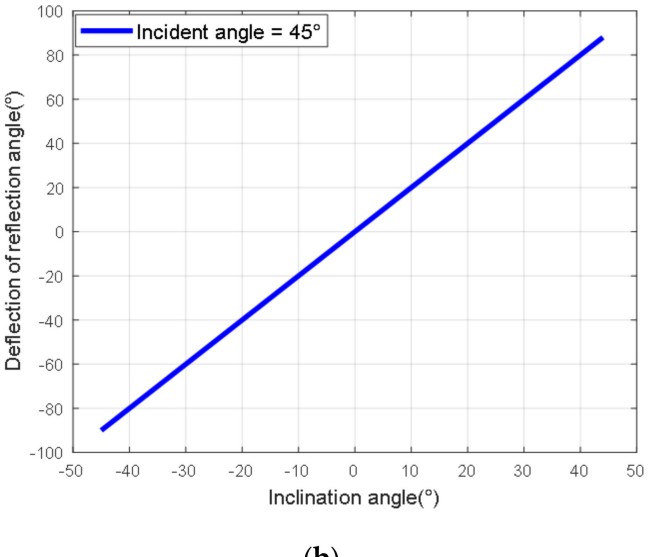

(**b**)

**Figure 8.** (**a**) Deflection of the reflection angle $\psi$ as a function of the inclination of the WAI when $\theta_{\text{pro}} = 30°$. (**b**) Deflection of the reflection angle $\psi$ as a function of the inclination of the WAI when $\theta_{\text{pro}} = 45°$.

*4.2. Resolution Analysis*

In this paper, we perform quasi-periodic sampling for the WAI, utilizing the position information of the feature points of the structured light pattern, while the neighboring sampling interval is approximately equal to $D$ (Figure 9). Reducing $D$ enables the recovery of shorter WAI wavelengths. However, as we describe below, a shorter D decreases the angular resolution of $\hat{\mathbf{N}}$. The angular resolution of $\hat{\mathbf{N}}$ increases with $z_h$. Here, we further analyze the relationship between the angular resolution of $\hat{\mathbf{N}}$ with the interval $D$ and $z_h$ in the meridional plane, based on geometric optics.

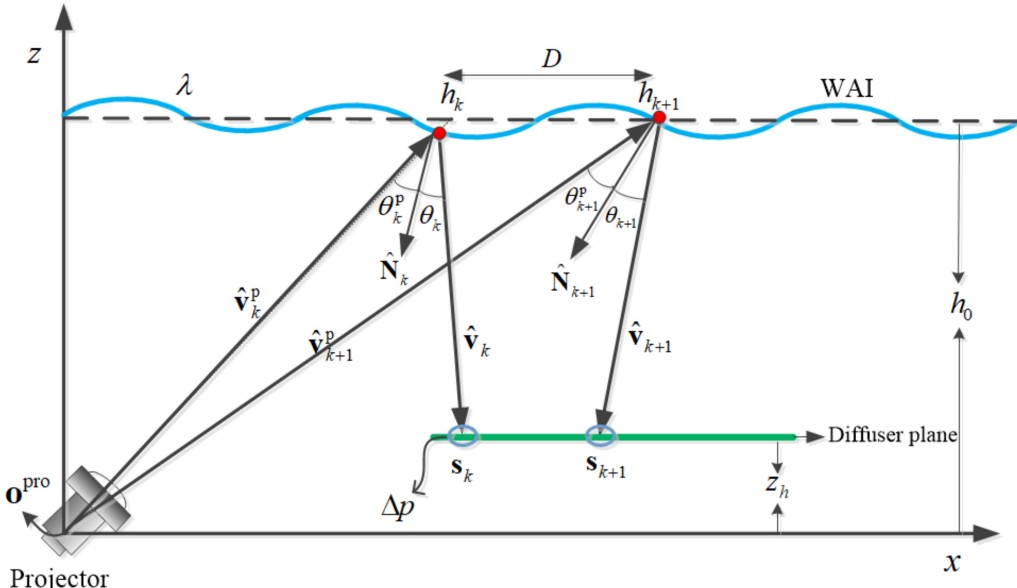

**Figure 9.** Geometry of the structured light projection system $S$.

Assume that the $x$ coordinate of the sampled point $\mathbf{q}_k$ is $h_k$, while $h_{k+1} - h_k \approx D$. The projected ray $\hat{\mathbf{v}}_k$ reflects by the WAI, and then irradiates a spot on the diffuser plane, at location ${}^Sk$. The reflected angle can be given by

$$\theta_k^{\mathrm{w}} = \arccos\left(\hat{\mathbf{v}}_k \cdot -\hat{\mathbf{z}}\right) = \theta_k^{\mathrm{flat}} + \psi_k = \arccos\left(\hat{\mathbf{v}}_k' \cdot -\hat{\mathbf{z}}\right) + \psi_k, \tag{25}$$

where $\psi_k$ is the angular deviation of the reflected ray (relative to the angle in flat water, $\theta_k^{\mathrm{flat}}$). ${}^Sk$ can be calculated as

$$s_k = h_k + (h_0 - z_h)\,\tan\theta_k^{\mathrm{w}}, \tag{26}$$

where $h_0$ is the average underwater depth of the projector. As the projector follows a central projection, the reflected angles of the adjacent sampling points $h_k$ and $h_{k+1}$ can be expressed as

$$\theta_k^{\mathrm{flat}} = \arctan\frac{h_k}{h_0}, \tag{27}$$

$$\theta_{k+1}^{\mathrm{flat}} = \arctan\frac{h_k + D}{h_0}. \tag{28}$$

Suppose that the projected ray $\hat{\mathbf{v}}_{k+1}$ corresponding to $h_{k+1}$ is parallel to the projected ray $\hat{\mathbf{v}}_k$ corresponding to $h_k$, and the corresponding location of the spot $s_{k+1}^0$ is

$$s_{k+1}^0 = h_{k+1} + \frac{h_0 - z_h}{h_0}\,h_k. \tag{29}$$

Actually, the location of the spot $s_{k+1}$ on the diffuser plane corresponding to $h_{k+1}$, according to the central projection, is given by

$$s_{k+1} = h_{k+1} + \frac{h_0 - z_h}{h_0}h_{k+1}. \tag{30}$$

From Equations (29) and (30), we obtain

$$\left\| s_{k+1}^0 \overrightarrow{\phantom{x}} s_{k+1} \right\| = \frac{h_0 - z_h}{h_0}D. \tag{31}$$

Let the sensor $S$ determine $s_k$ with spatial uncertainty $\Delta p$. This uncertainty may be due to the camera pixel size, diffuser characteristics and so on. This spatial uncertainty converts to uncertainties in the measured angular deviation $\Delta \psi$ and the measured angular deviation of the inclination angle of the WAI normal $\Delta \theta$,

$$\Delta p = \frac{h_0 - z_h}{\cos^2 \theta_k^{\mathrm{w}}} \Delta \psi \approx \frac{h_0 - z_h}{\cos^2 \theta_k^{\mathrm{flat}}} \frac{d\psi}{d\theta} \Delta \theta, \tag{32}$$

Given $\Delta p$, $z_h$, $h_0$ and $\Delta \theta$, when the WAI is flat, it is easy to establish the correspondence of $h_k$ with $s_k$, since $s_k < s_{k+1}$. To maintain correct correspondence, this relation has to be satisfied also when the WAI is wavy, $\forall_k$:

$$s_k + \Delta p/2 < s_{k+1}, \tag{33}$$

Equation (33) includes the small margin $\Delta p/2$, which guarantees that the adjacent projected spots $s_k$, $s_{k+1}$ do not merge into a single spot. From Equations (25), (26), (32) and (33),

$$
\begin{aligned}
D + \frac{h_0 - z_h}{h_0} D - \frac{\Delta p}{2} \quad &> (h_0 - z_h)(\tan \theta_k^{\mathrm{w}} - \tan \theta_{k+1}^{\mathrm{w}}) \\
&\approx \frac{h_0 - z_h}{\cos^2 \theta_k^{\mathrm{flat}}} (\psi_k - \psi_{k+1}) \\
&\approx \frac{h_0 - z_h}{\cos^2 \theta_k^{\mathrm{flat}}} \frac{d\psi}{d\theta} (\theta_k - \theta_{k+1})
\end{aligned} \tag{34}
$$

Therefore, when $h_0 - z_h$ is too large or too small, $D$ limits the dynamic range of the WAI slope changes $[\theta_k - \theta_{k+1}]$; beyond this range, it will cause a spot detection error.

Assume that the WAI has spatial period $\lambda$, in which the WAI inclination varies as $\theta(x) = \Phi \cos(2\pi x/\lambda)$, and $\Phi > 0$. In the worst case, at which $s_{k+1} - s_k$ is minimal, the WAI angle changes maximally between samples: $|\theta_k - \theta_{k+1}| = 2\Phi$. To avoid correspondence errors, the sampled interval $D$ must satisfy

$$D > \frac{h_0}{2h_0 - z_h} \left( \frac{\Delta p}{2} + \frac{h_0 - z_h}{\cos^2 \theta_k^{\mathrm{flat}}} \frac{d\psi}{d\theta} 2\Phi \right) \approx \frac{h_0}{2h_0 - z_h} \left( \frac{\Delta p}{2} + 2\Phi \frac{\Delta p}{\Delta \theta} \right). \tag{35}$$

As described in Section 3.2, the WAI sampling is quasi-periodic when the WAI is wavy, while the sampling tends to be periodic whose period is $D$ around when the perturbation amplitude of the water surface becomes smaller. According to the Nyquist sampling criterion $(\lambda/2) > D$, combining it with Equation (35) yields

$$\frac{2h_0 - z_h}{h_0} \lambda > \frac{\Delta p}{2} \frac{\Phi}{\Delta \theta} \left( 8 + \frac{2\Delta \theta}{\Phi} \right), \tag{36}$$

where $\Delta \theta/\Phi$ represents the relative angular resolution in which the WAI can be recovered: the ratio of uncertainty to dynamic range. Since $2\Delta \theta/\Phi \ll 8$, we obtain an uncertainty principle

$$\frac{2h_0 - z_h}{h_0} \lambda > \frac{\Delta p}{2} \frac{\Phi}{\Delta \theta} \left( 8 + \frac{2\Delta \theta}{\Phi} \right). \tag{37}$$

As Equation (37) shows, the relative WAI slope angular resolution can be traded off for the spatial resolution $(\lambda)$ of the WAI, before errors stemming from aliasing and correspondence take effect.

## 5. Results

In the experiment, the proposed method was implemented in the MATLAB environment (MathWorks Co., Natick, MA, USA). This section first demonstrates the process of the WAI reconstruction and the image restoration. To verify the performance of the proposed method, we make a comparison with the state-of-the-art method, namely Alterman's method [23], whose application scenarios and scopes are similar to those of our method.

We tested the two methods with the same data set. The source codes and test data are available online in [39].

*5.1. System Parameters*

In this paper, we coded our image recovery scheme described above, which can simulate the process of WAI reconstruction and image restoration according to the system parameters. The system parameters including the projector, camera v for observing airborne scenes and camera s for capturing the structured light image are shown in Table 1. Moreover, the following parameters are included: $h_0 = 150$mm, $\theta_{pro} = 40°$, $z_h = 60$mm.

**Table 1.** System parameters of the projector, camera v and camera s.

| System Parameters | Projector | Camera v | Camera s |
|---|---|---|---|
| CCD/LCD size | $4.1 \times 4.1$ mm | $5.0 \times 4.0$ mm | $5.0 \times 4.0$ mm |
| Image resolution | $1000 \times 1000$ | $800 \times 600$ | $800 \times 600$ |
| $f$ | 4.2 mm | 3.0 mm | 2.0 mm |
| Rotation matrix | $\begin{bmatrix} 0.6428 & 0 & -0.7660 \\ 0 & 1 & 0 \\ 0.7660 & 0 & 0.6428 \end{bmatrix}$ | $\begin{bmatrix} 0.8988 & 0 & -0.4384 \\ 0 & 1 & 0 \\ 0.4384 & 0 & 0.8988 \end{bmatrix}$ | $\begin{bmatrix} 1 & 0 & 0 \\ 0 & 1 & 0 \\ 0 & 0 & 1 \end{bmatrix}$ |
| Translation vector | $\begin{bmatrix} 0 & 0 & 0 \end{bmatrix}$ | $\begin{bmatrix} -96.17 & 0 & -46.90 \end{bmatrix}$ | $\begin{bmatrix} -286.40 & 0 & 190 \end{bmatrix}$ |

*5.2. WAI Reconstruction*

5.2.1. WAI Simulation

The motion of ocean waves is a complex random process. Using a spectrum to describe ocean waves is one of the most effective means to study ocean waves, since the modeling process of the spectrum is based on a large number of actual observation data [40–43]. According to the Longuet–Higgins model [40], the height distribution of ocean waves can be expressed as

$$\eta(x, y, t) = \sum_{i=1}^{M} \sum_{j=1}^{N} a_{ij} \cos[\omega_i t - k_i(x \cos\theta_j + y \sin\theta_j) + \varepsilon_{ij}], \tag{38}$$

where $\eta(x, y, t)$ represents the height distribution of a point $(x, y)$ on the WAI at time $t$, $a_{ij}$ is the amplitude of each harmonic, $\omega_i$ is the harmonic frequency, $k_i$ is the wave number of the harmonic, $\theta_j$ is the azimuth of the harmonic, $\varepsilon_{ij}$ is the initial phase of the harmonic, and $M$, $N$ represent the sampling number of the frequency range of the wave spectrum and the sampling number of the azimuth angle, respectively.

The amplitude of the harmonics $a_{ij}$ can be expressed in the wave spectrum as

$$a_{ij} = \sqrt{2S(\omega_i, \theta_j)d\omega d\theta} = \sqrt{2S(\omega_i)\varphi(\theta_j)\Delta\omega\Delta\theta}, \tag{39}$$

where $S(\omega, \theta)$ is the direction spectrum. $S(\omega)$ and $\varphi(\theta)$ denote the spectrum and directional distribution functions of ocean waves, respectively. $\Delta\omega$ and $\Delta\theta$ are the sampling interval for frequency and direction angle, respectively. According to the linear wave theory [44], $k_i$ and $\omega_i$ meet the dispersion equation

$$k_i = \frac{\omega_i^2}{g}, \tag{40}$$

where $g$ is the acceleration of gravity.

There are various versions of the directional spectrum. We use the P-M spectrum and the directional distribution function suggested by ITTC (International Towing Tank Conference), whose expression is as follows [45]:

$$S(\omega, \theta) = S(\omega)\varphi(\theta), \tag{41}$$

$$S(\omega) = \frac{8.1 \times 10^{-3} \times g^2}{\omega^5} \exp\left[-0.74 \times \left(\frac{g}{U\omega}\right)^4\right], \tag{42}$$

$$\varphi(\theta) = \frac{2}{\pi}\cos^2(\theta), \tag{43}$$

where $U$ is the average wind speed at a height of 19.5 m above the sea surface. In Figure 10, we show the simulated ocean wave when $U = 1.0$ m/s, $t = 10$ s.

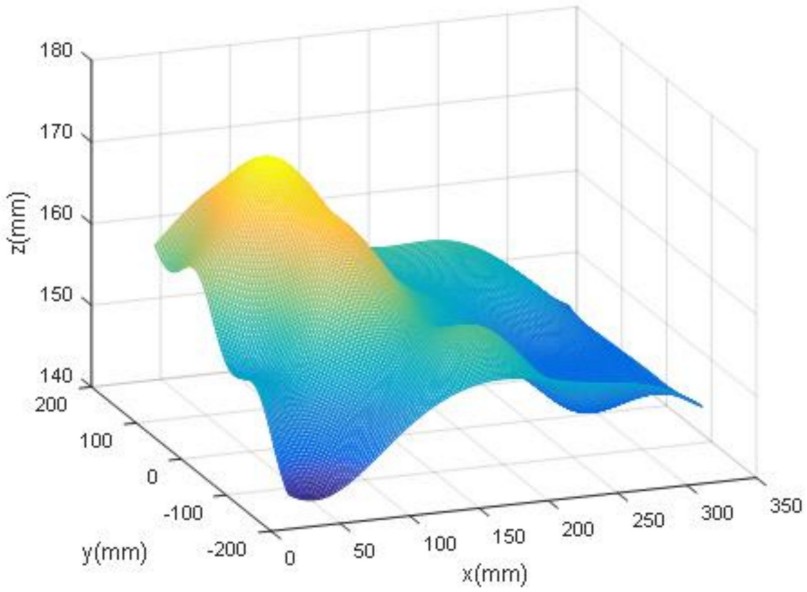

**Figure 10.** Simulated sea surface waves ($U = 1.0$ m/s, $t = 10$ s ).

5.2.2. Reconstruction of the WAI

An example of a full-system computer simulation is shown in Figure 11. A distorted structured light image was captured by camera s from the diffuser plane. The structured pattern projected by the projector is reflected by the WAI (Figure 10) and then forms the distorted structured light on the diffuser plane. Using the locations of feature points of the distorted structured light image and reference image, the WAI shape is estimated based on the finite difference method.

Intuitively, the reconstructed WAI is similar to the ground truth, but it exhibits a bias, as shown in Figure 11f. We use the root mean square error (RMSE) as the objective evaluation index. RMSE is given by

$$\text{RMSE} = \sqrt{\frac{1}{RC}(\sum_{i=1}^{R}\sum_{j=1}^{C}\eta(x_i, y_j) - \overline{\eta})}, \tag{44}$$

From Equation (44), the value of RMSE is 1.7845 mm. The absolute error distribution between the reconstructed WAI and the ground truth is shown in Figure 12. The unrecoverable bias is explained in Section 3.2.1. The research results show that the algorithm can reconstruct the wave surface when the error is allowed.

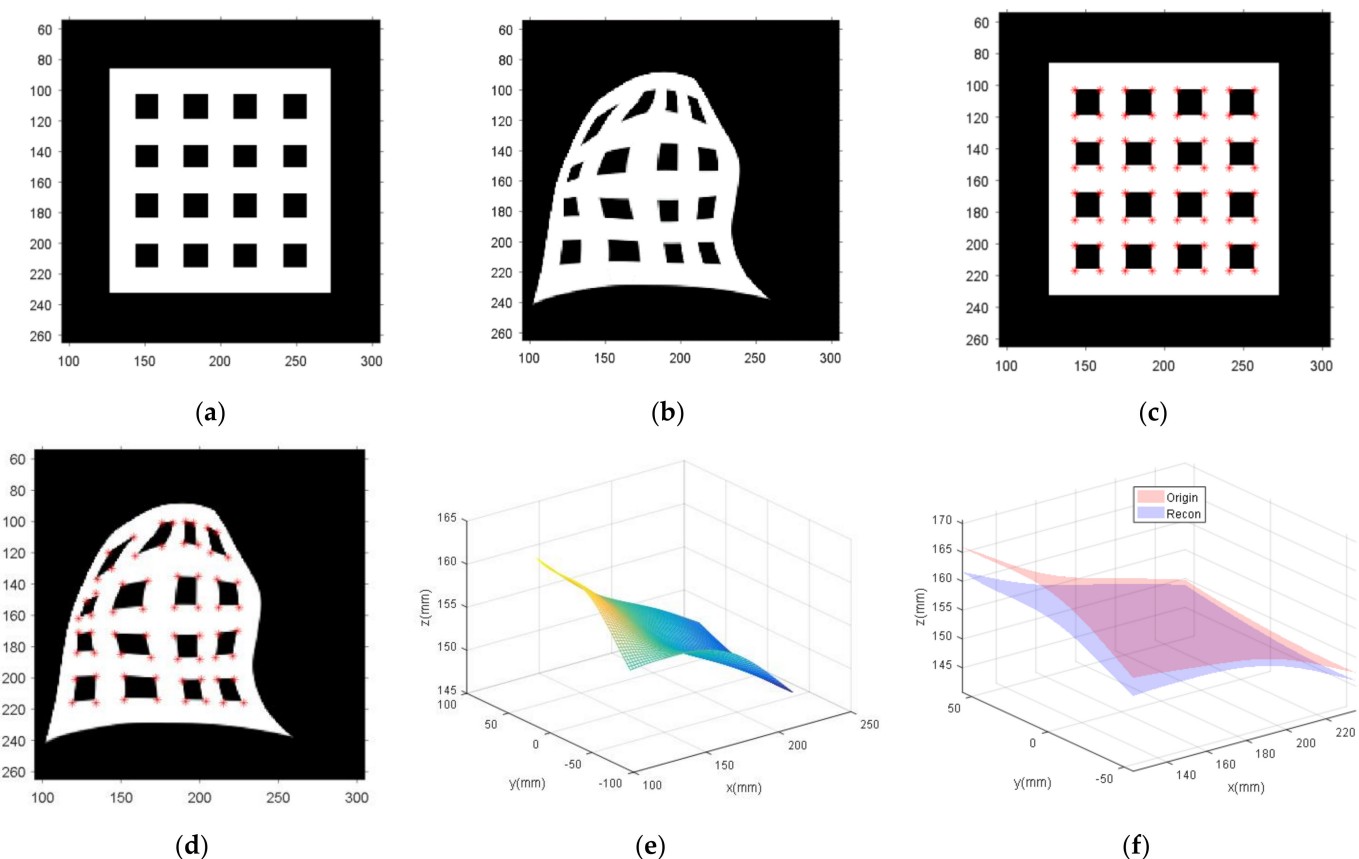

**Figure 11.** Simulation for WAI shape reconstruction. (**a**) Reference structured light image. (**b**) Distorted structured light image. (**c**) Results of feature point extraction for the reference structure light image using the method of Ref. [34]. (**d**) Results of feature point extraction for the distorted structured light image using the method of Ref. [34]. (**e**) Recovered WAI shape. (**f**) Ground-truth WAI [red] and reconstructed WAI [blue].

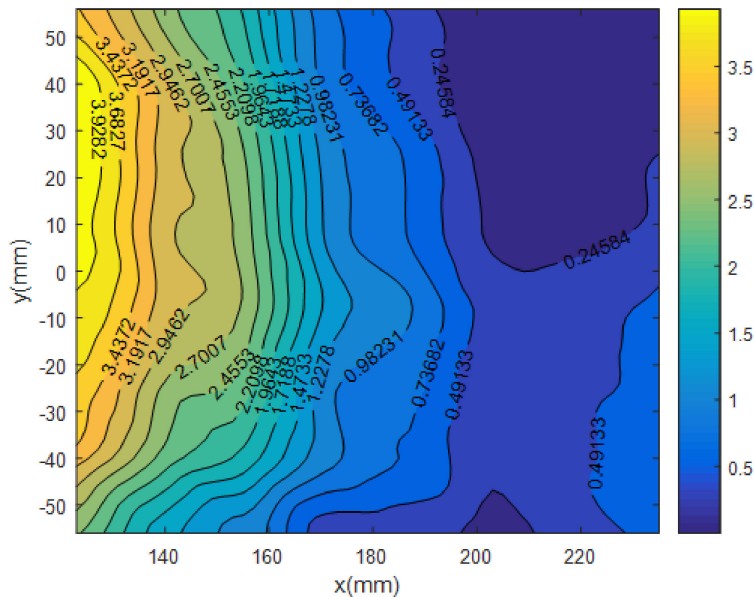

**Figure 12.** Absolute error distribution between the reconstructed WAI and the ground-truth WAI.

### 5.2.3. Comparative Analysis with Alterman's Method

Alterman's method [23] has strict requirements for natural illumination, is not suitable for cloudy weather and requires sunlight. Moreover, the accuracy of WAI reconstruction varies with the position of the sun. Here, we adopt Alterman's method and our algorithm to reconstruct the same WAI (Figure 10), respectively. The relationship between the absolute error distribution of the reconstructed WAI and the ground truth with the incident angle of the sunray $\theta_{sun}$ using Alterman's method is shown in Figure 13. Figure 11 shows the RMSE and the maximum absolute error ($Ae_{max}$) as a function of the zenith angle of the sunray using Alterman's method. Combining Figures 13 and 14, it is obvious that the restoration accuracy is limited by the illumination conditions. As the zenith angle of the sunray increases, the maximum absolute error and the stability of the system decrease. Furthermore, as shown in Figure 14, the minimum mean square error of Alterman's method is 3.4603 mm, while the RMSE of our algorithm is 1.7845 mm.

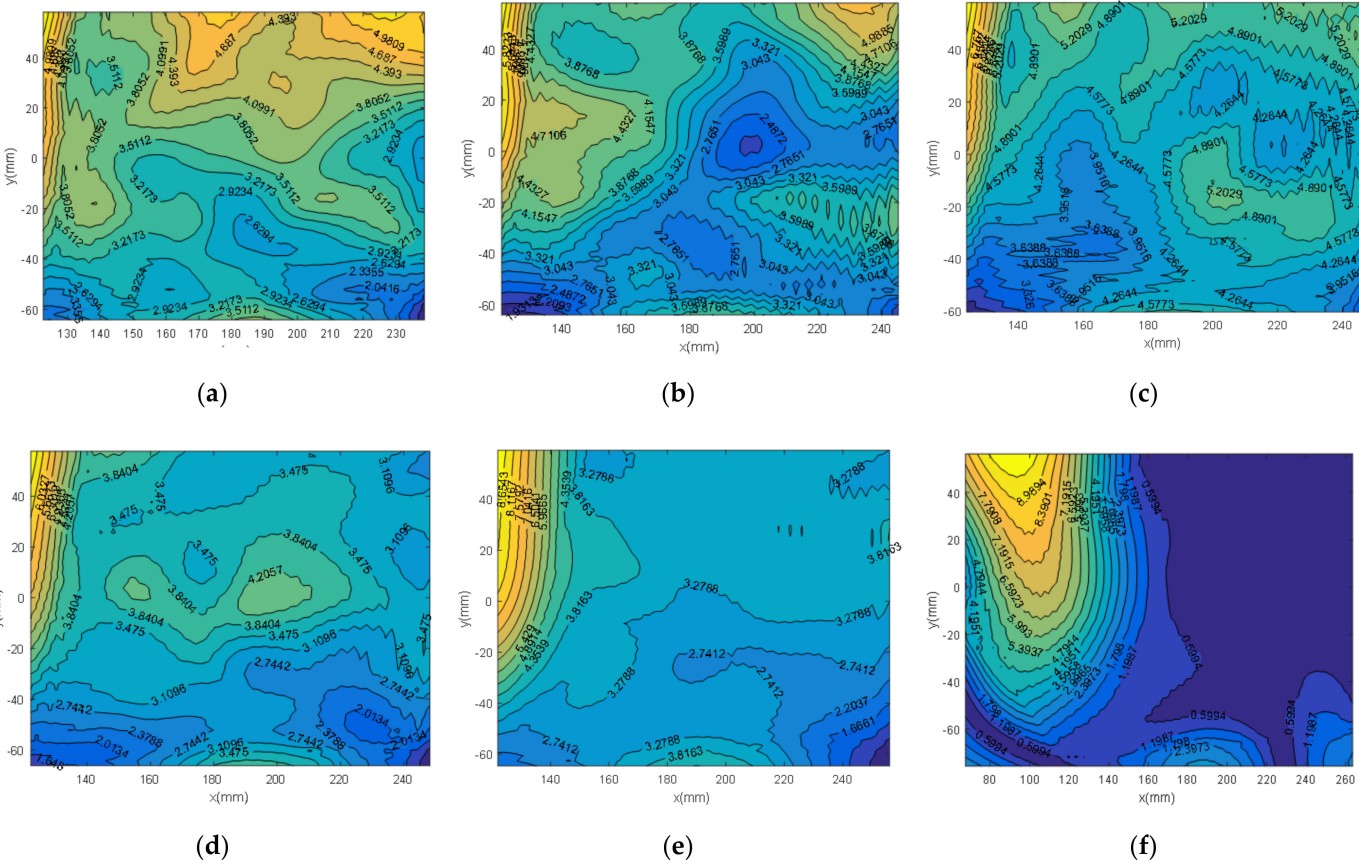

**Figure 13.** Absolute error distribution between reconstructed WAI and the ground truth for the different $\theta_{sun}$ using Alterman's method. (**a**) Absolute error distribution when $\theta_{sun} = 0°$; (**b**) absolute error distribution when $\theta_{sun} = 15°$; (**c**) absolute error distribution when $\theta_{sun} = 30°$; (**d**) absolute error distribution when $\theta_{sun} = 45°$; (**e**) absolute error distribution when $\theta_{sun} = 60°$; (**f**) absolute error distribution when $\theta_{sun} = 75°$.

The research results show that, compared with the model included in Alterman's method, the proposed algorithm can overcome the influence of changes in natural illumination conditions for WAI reconstruction and improve the accuracy of WAI restoration.

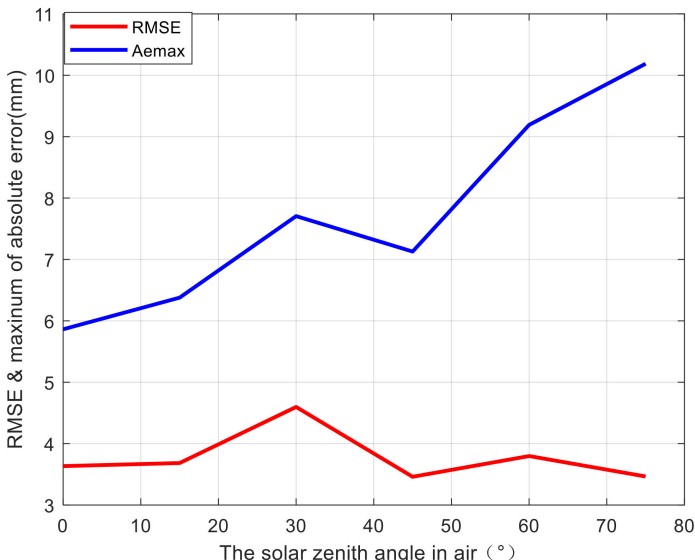

**Figure 14.** The RMSE and the maximum absolute error ($Ae_{max}$) as a function of the zenith angle of the sunray using the method in Ref. [23].

### 5.3. Image Restoration

The submerged camera images a checkerboard located at height $z_a = h_0 + 1000$ through the wavy WAI. The results are shown in Figure 15; the size of the image is $300 \times 250$ pixel. The scatter plot shows the coordinates of the corner points of the color-coded square checkerboards in the following three images, namely the ground truth (red), the distorted image (purple) and the recovery image (blue). In addition, the standard deviation (STD) of the corner positions in the distorted image is 26.2783 pixels, and the STD of the image restored by the estimated WAI is reduced to 1.2247 pixels. The results show that the image restoration method via structured light projection can significantly reduce the distortion.

#### 5.3.1. Image Quality Metrics

We use three standard image quality/similarity metrics for quantitative evaluation: (1) mean square error (MSE) [2], (2) peak signal-to-noise ratio (PSNR) [46], (3) structural similarity index (SSIM) [47], in which the expressions of MSE, PSNR and SSIM are, respectively,

$$\text{MSE} = \frac{\sum\limits_{i=1}^{w} \sum\limits_{j=1}^{h} [F(i,j) - I(i,j)]^2}{w \times h}, \tag{45}$$

$$\text{PSNR} = 10 \log_{10} \left( \frac{\max(I)^2}{\text{MSE}} \right), \tag{46}$$

$$\text{SSIM} = \frac{(2u_F u_I + c_1)(2\sigma_{FI} + c_2)}{(u_F^2 + u_I^2 + c_1)(\sigma_F^2 + \sigma_I^2 + c_2)}. \tag{47}$$

where $I$ is the ground-truth image. $\max(\cdot)$ denotes the maximum possible value. $u_F$ and $u_I$ represent the means of the image $F$ and the image $I$, respectively. $\sigma_F^2$ and $\sigma_I^2$ represent the variance of $F$ and $I$, respectively. $\sigma_{FI}$ is the covariance of $F$ and $I$. $c_1 = (0.01 \times L)$ and $c_2 = (0.03 \times L)$ are constants, in which L represents the dynamic range of pixel values.

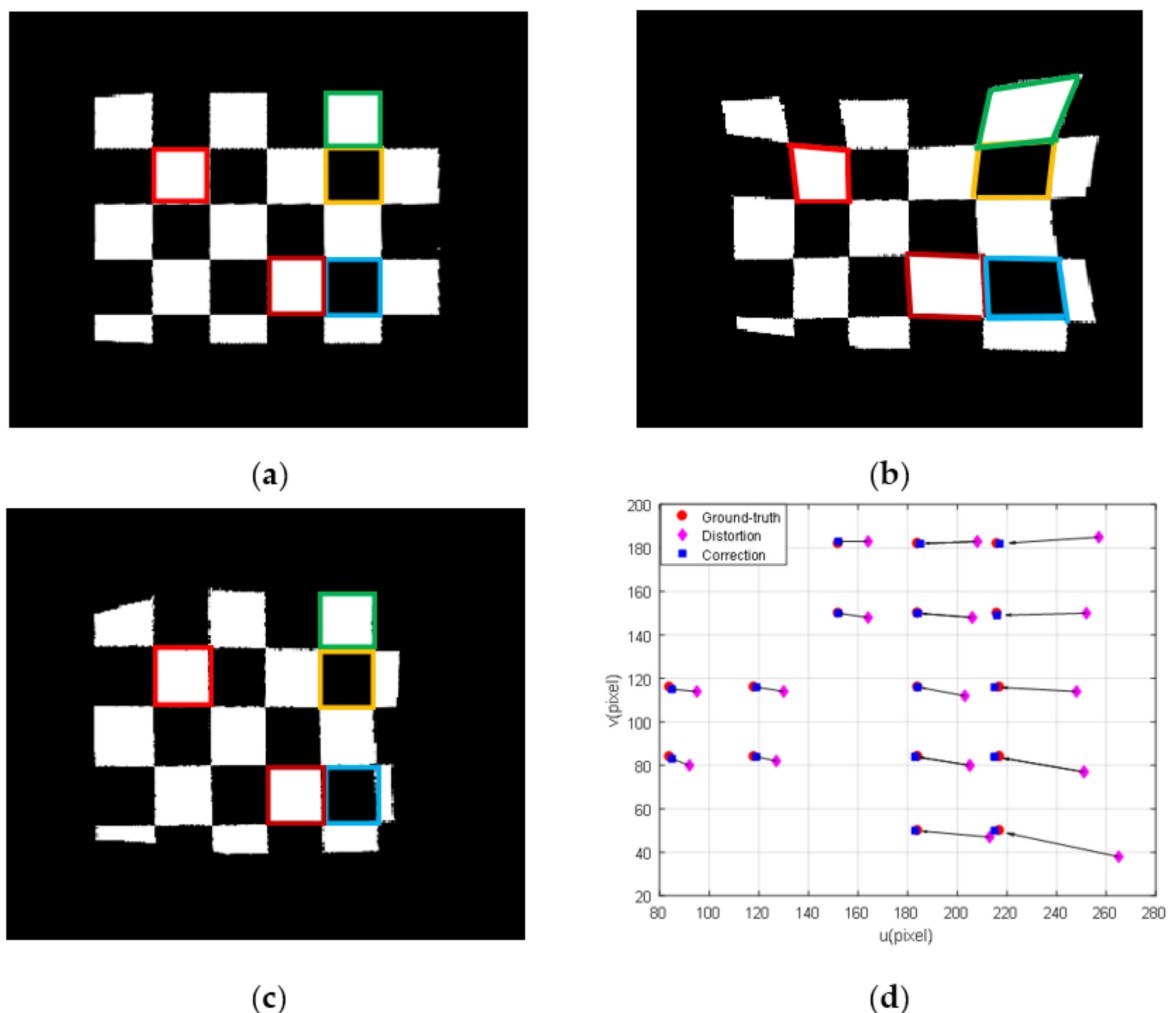

**Figure 15.** (**a**) Ground-truth image; (**b**) an image distorted by a wavy WAI; (**c**) recovered image; (**d**) a scatter plot showing coordinates of the corner points of the color-coded square checkerboards in above three images.

5.3.2. Results of Quantitative Analysis

Figure 16 shows the results of the two methods to recover the distorted image, and the comparison results are presented in Table 2. The results show that compared with Alterman's method, the proposed method has a significant improvement in performance indicators such as PSNR, MSE and SSIM, which proves the effectiveness of the algorithm.

**Table 2.** Comparison of image quality metrics among the proposed method, the distorted image and Alterman's method [23].

| | SSIM (H) | | | MSE (L) | | | PSNR (H) | | |
|---|---|---|---|---|---|---|---|---|---|
| | Data 1 | Data 2 | Data 3 | Data 1 | Data 2 | Data 3 | Data 1 | Data 2 | Data 3 |
| Distortion | 0.5584 | 0.6558 | 0.6140 | 0.1367 | 0.0653 | 0.0914 | 8.6414 | 11.8538 | 10.3925 |
| Alterman [23] | 0.6814 | 0.6784 | 0.6270 | 0.0520 | 0.0441 | 0.0514 | 12.8389 | 13.5571 | 12.8878 |
| Proposed method | 0.7630 | 0.7877 | 0.7434 | 0.0317 | 0.0461 | 0.0297 | 15.00 | 13.4651 | 15.2656 |

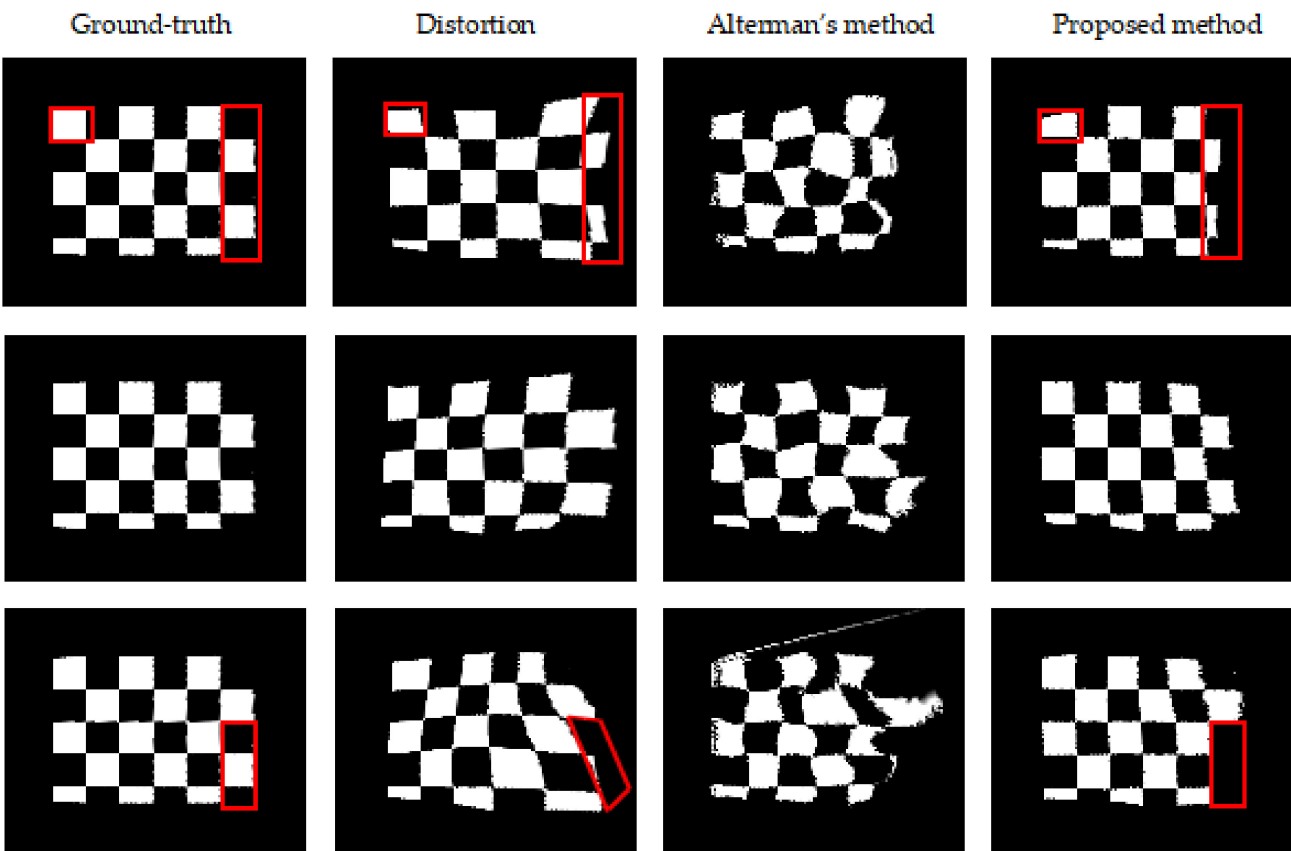

**Figure 16.** Results of images recovered by the estimated wavy WAI with the two methods [23].

## 6. Conclusions

The image restoration model via structured light projection is a novel approach for virtual periscopes. Different from previous methods, we do not require special natural illumination [15,16,23,25], multiple viewpoints [24] or image accumulation processes [17–22]; we only require a simple projection setup and an image of the distorted scene. This means that our method can be applicable to more scenarios—for instance, monitoring the habits of seabirds, path planning and obstacle avoidance for underwater vehicles, airborne target detection, recognition and tracking and seafloor mapping, etc.

In Section 5, we coded our image recovering scheme described above, which can simulate the process of WAI reconstruction and image restoration according to the system parameters. Compared with the state-of-the-art method, our method can overcome the influence of changes in natural illumination conditions for WAI reconstruction and improve the accuracy of WAI restoration. Furthermore, the results show that our method can significantly reduce the distortion and performs better in the recovery of distortions. In the future, we plan to conduct further tests in the laboratory and at sea to verify the effectiveness of our approach.

Similar to other instantaneous distorted image restoration algorithms [11,15,17,23,25], the proposed method still cannot eliminate the problem of loss of image details caused by random refraction of the WAI (marked in red in Figure 16). However, this problem can later be handled by image fusion algorithms [48–51] after images are corrected by our process. Rather than addressing the full-blown distortions in raw images, such video post-processing methods may handle more easily images whose distortions are residual.

In Section 4, the limitation of the algorithm is analyzed, and it is found that errors stemming from aliasing and correspondence will occur in structured light when the spatial frequency of the WAI is large. In the future, it is necessary to further explore a solution to this problem, such as structured light encoding [52], optical flow method [53], etc.

**Author Contributions:** All authors have made significant contributions to this paper. B.J. designed the research, conducted simulation analysis and wrote the manuscript; C.M. provided guidance; D.Z. conducted data curation; Y.S. conducted validation; J.A. provided supervision and funding. All authors have read and agreed to the published version of the manuscript.

**Funding:** This research was funded by the Guangxi National Science Foundation, grant number 2018GXNSFAA294056, and the Guangxi Young and Middle-Aged Teachers' Basic Research Ability Improvement Project (2022KY0703).

**Data Availability Statement:** All data or code used to support the findings of this study are available from the corresponding author.

**Conflicts of Interest:** The authors declare no conflict of interest.

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
