# Peer review of "Seeing through Wavy Water–Air Interface: A Restoration Model for Instantaneous Images Distorted by Surface Waves"

_futureinternet, doi:10.3390/fi14080236_

Round 1

Reviewer 1 Report

The authors proposed an image recovery procedure using blue-green structured light pattern. Their algorithm at the first step determines the structure of WAI and then tries to restore distorted image. The overall organization of the paper is fine. Geometrical analysis is carried out for shape determination. Finally, the results are compared to a conference paper published in 2014. The paper may be published after fully addressing following concerns:

1) Line 239: Please explain how normal vectors can be determined as [Z_x(x,y), Z_y(x,y), -1].

2) line 40,41: different font size compared to rest of manuscript.

3) It is recommended to use past or present perfect tenses in the literature review.

4) Only one algorithm is used as baseline (ref. 23). The authors must compare their work to some recent works.

5) There are a lot of errors in the manuscript. I have listed some of them. The English should be improved.

abstract: since light rays bent ==> since light rays are bent

In the first part, the algorithm ==> In the first part, an algorithm

Then we synchronously recovers ==> Then we synchronously recover

The experimental results show that , ==> The experimental results show that

compared with the state-of-art method ==> compared to the state-of-the-art methods

Line 32: The study shows ==> some studies show that (you cited two references)

line 103: rewrite this sentence: Imaging through water-air interface using a submerged camera, it can observe the whole sky

line 106: this because light rays bend when entering or exiting water ==> this happens because the light rays are bent when entering or exiting water

line 139: water.Studies ==> water. Studies

line 140: two unrelated physical processes ==> two independent physical processes

line 149: C.Smith ==> C. Smith

170: the algorithm for ==> an algorithm for

172: we synchronously recovers ==> we synchronously recover

Figure 4: S and V are indicated by capital fonts in the caption, however, in the text body small fonts are used. The authors must use consistent notation through the manuscript.

Line 185: There is a hyperlink for "take advantage of". I assume that authors copied this part from a translating website (https://fanyi.baidu.com/?aldtype=85&keyfrom=alading#en/zh/take%20advantage%20of). Make sure to paste "plain text" or "unformatted text" !

Line 189: when the water surface is still ==> the sentence is incomplete (Or if you mean stationary, then choose a better work to describe WAI)

line209: corresponding spot when the WAI is still. ==> the sentence is incomplete (Or if you mean stationary, then choose a better work to describe WAI)

line 214: deltah >> h_0 ==> deltah << h_0

line 259, 260: The Principle of 3D Camera Imaging ==> external hyperlink again (https://link.springer.com/chapter/10.1007/978-1-4614-7400-5_13)

line 364: implemented on MATLAB ==> implemented in MATLAB environment

Author Response

Dear Judges:
         First of all, thank you very much for reviewing my manuscript in your busy schedule. Your review comments and suggestions are very important to our work. For your review comments (2), (3), (5), we have carefully revised and reflected in the submitted revised manuscript. In addition, for the review comments (1) and (4), we will provide specific explanations in Annex 1.

         Finally, thank you very much for your excellent review work. I look forward to hearing from you if you have any questions.

Reviewer 2 Report

Dear Authors,

The manuscript entitled “Seeing through wavy water-air interface: a restoration model for the instantaneous images distorted by surface waves” seems to fit the aims of the scientific journal Future Internet. The title reflect the content and emphasize the paper's interest and significance. Regarding the conception of the paper and the presented results, the paper can be graded as good, but requires some improvements and correction before publishing.

To publish this paper, the following revisions are recommended:

·         Abstract/ The abstract does not fully explain the meaning of the paper. The suggestion to the authors is to be specific and precise in the abstract, to provide insight into the analyzes performed, as well as specific results.

·         Introduction/ Please improve introduction with review of the state-of-the-art in the paper. A detailed-enough discussion of similar experiences and similar tools should be added and clearly labelled as related works

·         Conclusions/ Given the scope of the results presented, it is suggested that this section be improved. Focus more on how your research contributed to gaps in knowledge; add the scientific and practical significance of the chosen method.

Author Response

Dear judge:

    First of all, thank you very much for reviewing my manuscript in your busy schedule. Your review comments and suggestions are very important to our work.

     For your review comments , I have revised your comments one by one, you can view the relevant parts in the revised manuscript, and I have marked the revised parts. I don't know if the result of my modification will satisfy you.                  Finally, thank you very much for your excellent review work and look forward to your reply.

     Sincerely Yours

     Bijian Jian

Reviewer 3 Report

The work needs the following corrections:

- it is worth expanding the motivation and slightly information about what is the greatest achievement of this work in relation to other similar solutions,

- it is necessary to add to the work codes with implementations of particular issues,

-it is worth thinking about expanding the experimental part with a broader verification of the usefulness of the proposed method. In particular, practical tests on some real set of objects in the context of pattern recognition.

- One might also be tempted to lighten the math part a bit, with visualizations of the issues shown - where possible.

- finally, I believe that this work has research potential, after expanding the experimental part will be a valuable item.

Author Response

Dear reviewer

   First of all, thank you very much for reviewing my manuscript in your busy schedule. Your review comments and suggestions are very important to our work.

     According to your review comments, I have revised the manuscript one by one. You can check the relevant parts in the revised manuscript. I have marked the revised parts. I don't know if the result of my modification will satisfy you.

   Moreover, for your comment [3], we will provide specific  explanation  i in the attachment.

   Finally, thank you very much for your excellent review work. I look forward to hearing from you if you have any questions.

Round 2

Reviewer 3 Report

I have no further comments on this work. The work takes into account my comments.